

# Phylogenetic and recombination analyses of two deformed wing virus strains from different honeybee species in China

Dongliang Fei[1,2,*], Yaxi Guo[1,*], Qiong Fan[3], Haoqi Wang[1], Jiadi Wu[1], Ming Li[1] and Mingxiao Ma[1]

[1] Institute of Biological Sciences, Jinzhou Medical University, Jinzhou, Liaoning, China
[2] College of Veterinary Medicine, Northeast Agricultural University, Haerbin, Heilongjiang, China
[3] Jinzhou Animal Disease Prevention and Control Center, Jinzhou, Liaoning, China
* These authors contributed equally to this work.

Corresponding author
Mingxiao Ma,
mamingxiao@jzmu.edu.cn

## ABSTRACT

**Background:** Deformed wing virus (DWV) is one of many viruses that infect honeybees and has been extensively studied because of its close association with honeybee colony collapse that is induced by *Varroa destructor*. However, virus genotypes, sequence characteristics, and genetic variations of DWV remain unknown in China.

**Methods:** Two DWV strains were isolated from Jinzhou and Qinhuangdao cities in China, and were named China1-2017 (accession number: MF770715) and China2-2018 (accession number: MH165180), respectively, and their complete genome sequences were analyzed. To investigate the phylogenetic relationships of the DWV isolates, a phylogenetic tree of the complete open reading frame (ORF), structural protein VP1, and non-structural protein 3C+RdRp of the DWV sequences was constructed using the MEGA 5.0 software program. Then, the similarity and recombinant events of the DWV isolated strains were analyzed using recombination detection program (RDP4) software and genetic algorithm for recombination detection (GARD).

**Results:** The complete genomic analysis showed that the genomes of the China1-2017 and China2-2018 DWV strains consisted of 10,141 base pairs (bp) and 10,105 bp, respectively, and contained a single, large ORF (China1-2017: 1,146–9,827 bp; China2-2018: 1,351–9,816 bp) that encoded 2,894 amino acids. The sequences were compared with 20 previously reported DWV sequences from different countries and with sequences of two closely related viruses, Kakugo virus (KV) and *V. destructor* virus-1. Multiple sequence comparisons revealed a nucleotide identity of 84.3–96.7%, and identity of 94.7–98.6% in amino acids between the two isolate strains and 20 reference strains. The two novel isolates showed 96.7% nucleotide identity and 98.1% amino acid identity. The phylogenetic analyses showed that the two isolates belonged to DWV Type A and were closely related to the KV-2001 strain from Japan. Based on the RDP4 and GARD analyses, the recombination of the China2-2018 strain was located at the 4,266–7,507 nt region, with Korea I-2012 as an infer unknown parent and China-2017 as a minor parent, which spanned the entire helicase ORF. To the best of our knowledge, this is the first study to the complete sequence of DWV isolated from *Apis cerana* and the possible DWV recombination events in China. Our findings are important for further research of

the phylogenetic relationship of DWVs in China with DWV strains from other countries and also contribute to the understanding of virological properties of these complex DWV recombinants.

# BACKGROUND

The honeybee is one of the most important pollinators and plays a crucial role in agricultural ecology. However, over the last decades, honeybee populations have rapidly decreased, which has led to a pollination crisis that seriously threatens global agricultural production (*Martin, 2001*). This dramatic decline of honeybee colonies was suggested to be the result of interactions between parasites and pathogens, including viruses, fungi, mites, bacteria, microsporidia, and other pests. Among the effects of pathogens, viral diseases are considered a major threat to apiculture, and 12–20 kinds of single-stranded positive sense "picorna-like" RNA viruses (*Berényi et al., 2007*; *Baker & Schroeder, 2008*; *Reddy et al., 2013*) have been confirmed to infect honeybees. Among these viruses, the deformed wing virus (DWV) is the most important honeybee virus, causing colony collapse disorder (*Vanengelsdorp et al., 2009*) owing to an interaction effect with *Varroa destructor*. The global prevalence of DWV is the presumed driver of the substantial frequency of honeybee colony collapse; thus, DWV is regarded as the most destructive honeybee virus infecting *Apis mellifera* and *A. cerana*, thereby threatening food safety and the equilibrium of various ecosystems.

Deformed wing virus is a member of the picorna-like insect virus family *Iflaviridae* and consists of a 30 nm icosahedral particle with a single positive strand RNA genome. This genome is about approximately 10 kb in length with a large open reading frame (ORF), which encodes a 2,894 amino acid polyprotein. After cleavage by viral proteases, polyproteins produce structural and non-structural proteins, and the major structural proteins are composed of four proteins, VP1, VP2, VP3, and VP4. The non-structural proteins include a helicase, a genome-linked viral protein, a 3C-protease (3C-pro), and an RNA-dependent RNA polymerase (RdRp). Furthermore, similar to other *Iflaviruses*, DWV structural and non-structural proteins are at the N-terminal end and C-terminal end of the polyprotein, respectively (*Lanzi et al., 2006*; *Berényi et al., 2007*; *De Miranda & Gensch, 2010*).

The clinical symptoms of DWV infection are atrophy of the wings, smaller body size, discoloration and paralysis in adult bees, and a generally shortened life span (*Kovac & Crailsheim, 1988*; *Prisco et al., 2011*). Although DWV infections in adult bees often produce no clinical symptoms, it is a serious threat to honeybee colonies (*Martin, 2001*). The virus was originally isolated from infected adult bees in Japan in the 1980s, and since then has been distributed globally, including throughout Asia, Europe, Africa, North America, South America, and the Middle East (*Bailey, Carpenter & Woods, 1981*; *Kovac & Crailsheim, 1988*; *Calderon et al., 2003*; *Ellis & Munn, 2005*). DWVs can be categorized into two master variant strains, DWV-A and DWV-B (*Moore et al., 2011*; *Mcmahon et al., 2016*; *Mordecai et al., 2016*). DWV-A comprises the original DWV strain and the Kakugo

virus (KV) (*Mcmahon et al., 2016*). *V. destructor* virus-1 (VDV-1) was classified as DWV-B, with an overall RNA genome of 84% to those of the classic DWV-A. In addition, the first 1,455 nt of the ORF encoding the lower molecular mass structural proteins shows the greatest diversion from those of the classic DWV-A, with an RNA identity of 79%, and translates to a polypeptide of 485 aa with an identity of 90% (*Ongus et al., 2004*; *Ryabov et al., 2014*). Recent studies have shown that recombination events have frequently occurred between these two main variants (*Mordecai et al., 2015*; *Zioni, Soroker & Chejanovsky, 2011*). Thus far, the complete genome sequences of approximately 20 strains have been sequenced, with isolates mostly obtained from *A. mellifera* and *Vespa crabro*; however, there are no complete genome sequences of DWV isolates from *A. cerana* (*Reddy et al., 2013*; *Lamp et al., 2016*; *Forzan et al., 2017*). The epidemiology of DWV in *A. cerana* and *A. mellifera* has been investigated in China; however, virus genotypes, sequence characteristics and genetic variations of DWV remain unknown (*Zheng et al., 2015*; *Chao et al., 2017*).

In the present study, we isolated two DWV strains (China1-2017 and China2-2018) from *A. mellifera* and *A. cerana*, respectively, and produced the complete nucleotide sequences of DWV from *A. cerana*. The sequences of the China1-2017 and China2-2018 isolates were analyzed and compared to the reference nucleotide sequences of DWV genotypes from other countries. Furthermore, we analyzed molecular biological characteristics and phylogenetic relationship of DWV structural and non-structural polyprotein regions of the China1-2017 and China2-2018 strains and reference strains. Moreover, we performed a recombination analysis of DWV from Chinese isolates using recombination detection program (RDP4) software and a genetic algorithm for recombination detection (GARD).

## METHODS

### Ethics statement and legal agreement

This research was approved by the Experimental Animal Ethics Committee of Jinzhou Medical University (No. 20180016) and the Agricultural and Rural Comprehensive Service Center of Jinzhou (No. 20180619).

### Sample collection

Samples were collected from two different regions in China. A total of 257 *A. mellifera* samples originating form Jinzhou of Liaoning Province (41°14′34″N, 121°8′15″E) were collected in March 2016, and 463 *A. cerana* worker bee samples originating from Qinhuangdao of Hebei Province (40°3′26″N, 119°33′4″E) were collected in April 2017. All sampled individuals exhibited typical deformed wing symptoms. Upon collection, the bees were stored on ice and immediately transported to the laboratory, where they were kept frozen at −80 °C until analysis.

### Virus isolation

Virus isolation was performed according to previously published methods (*Ying et al., 2016*; *Mingxiao et al., 2011*; *Jakubowska et al., 2016*) Briefly, 50–60 adult bees were

completely homogenized with a mortar and pestle in 10 mL of phosphate-buffered saline (pH 7.4), containing 0.5% nonylphenol ethoxylate, then the homogenate was incubated for 30 min at 20 °C. After this, the homogenate was centrifuged at 5,000×*g* for 20 min at 4 °C. Large debris was removed, and the supernatant was again centrifuged at 8,000×*g* for 30 min at 4 °C. After discarding the precipitate, the supernatant was placed in an ultracentrifuge tube and centrifuged at 82,000×*g* for 1 h at 4 °C. The resulting pellet was resuspended in two mL STE buffer (10 mM Tris-HCl, 0.1M NaCl, and one mM EDTA; pH 7.3). The viral strains of *A. mellifera* and *A. cerana* were named China1-2017 and China2-2018, respectively.

## Viral RNA isolation and DWV screening by PCR

Total RNA was extracted from the purified virus of China1-2017 and China2-2018 strains using a TIANamp Virus DNA/RNA Kit (Tiangen, Beijing, China), according to the manufacturer's instructions. Total RNA was eluted in 30 μL of diethyl pyrocarbonate-treated water and stored at −80 °C until further analyses. To determine the presence of DWV in *A. mellifera* and *A. cerana*, a RT-PCR assay was performed using a PrimeScript™ RT-PCR Kit (TaKaRa, Dalian, China), according to the manufacturer's instructions.

The primers DWV-F (5′-TTTGCAAGATGCTGTATGTGG-3′) and DWV-R (5′-GTCGTGCAGCTCGATAGGAT-3′) were used to amplify a 395 base pairs (bp) fragment of the DWV RdRp gene (accession number: AY292384). The PCR amplification was carried out under the following conditions: 94 °C for 2 min, followed by 25 cycles of 94 °C for 30 s, 55 °C for 30 s, 72 °C for 30 s, and a final extension of 72 °C for 5 min. A sample of six μL of PCR product was loaded on a 1.2% agarose gel containing Gelstain and analyzed using a Tanon 2500 Digital Gel Image Analysis System. The PCR products were sequenced commercially (Sangon Biotech, Shanghai, China).

## RT-PCR amplification and genome sequencing

Viral genomic RNA of China1-2017 and China2-2018 were reverse-transcribed to cDNA with an oligo (dT) primer (*Sambrook & Russell, 2001*) (TransGen Biotech, Beijing, China), according to the manufacturer's recommendations. A total of 17 primer pairs were designed to amplify the complete genome sequence of China1-2017 and China2-2018, using the complete DWV sequence of the USA and KOR strains (accession number: AY292384 and JX878304; Table 1) (*Lanzi et al., 2006*; *Reddy et al., 2013*). PCR amplifications were performed in Eppendorf tubes, and the cycling protocol for RT-PCR amplification was as follows: 45 min at 42 °C (reverse transcription), followed by 30 cycles at 95 °C for 60 s, 50–55 °C for 30 s, and 72 °C for 60 s. The 3′ and 5′ termini of the China1-2017 and China2-2018 strains were obtained by employing a rapid amplification of cDNA ends (RACE) technique, using a SMARTer RACE 5′/3′ kit (Clontech, Beijing, China). PCR products were electrophoresed and purified, and subsequently sequenced commercially (Sangon Biotech, Shanghai, China). The nucleotide sequences of all fragments were assembled to compile the genomic sequences of the China1-2017 and China2-2018 strains using Lasergene software (DNASTAR), based on published complete DWV sequences (accession number: AY292384, JX878304, and JX878305).

**Table 1 Primers designed for overlapping sequences to ensure the complete sequencing of the selected DWV genotypes.**

| Primer name | Sequence (5′–3′) | Nucleotide position | Amplicon size (bp) |
| --- | --- | --- | --- |
| DWV1F | TCCATAGCGAATTACGGTG | 8–26 | 741 |
| DWV1R | GTCCCAGCTCTATCGCAGAAA | 729–749 | |
| DWV2F | GAA GTG ACT AGC AAT CAT GGA | 605–625 | 716 |
| DWV2R | ATG TCG YCT GGT YAT AGA CG | 1320–1301 | |
| DWV3F | TCT GTY GCC YAT GCA CCT C | 1191–1210 | 691 |
| DWV3R | GCG CTG GAA TAG ATG TAC TAG | 1881–1861 | |
| DWV4F | ACC CTA ATC CAG GAC CTG AT | 1780–1799 | 657 |
| DWV4R | AGG TAG TTG GAC CAG TAG CAC | 2436–2416 | |
| DWV5F | AACAAGAATTGTGCCAGA | 2333–2350 | 716 |
| DWV5R | GTTGCAAAGATGCTGTCA | 3045–3028 | |
| DWV6F | CCG TGG GTG TAG TAT CTA G | 3011–3030 | 640 |
| DWV6R | GCG AGC TCG TTC AGC ATT AT | 3650–3631 | |
| DWV7F | AGC AAG CTG CTG TAG GAA CTC | 3547–3567 | 739 |
| DWV7R | TGA CCA GTA GAC ACA GCA TC | 4285–4266 | |
| DWV8F | ACA TCG ACC GGA TCG TAG A | 4211–4229 | 720 |
| DWV8R | AGT AAC CGC WTG ACT ACA GT | 4930–4911 | |
| DWV9F | GAA GAC AGT TGC TTG GGC GA | 4832–4851 | 796 |
| DWV9R | AGG AGT ACG ACT CGC ACG T | 5627–5609 | |
| DWV10F | GAT ATG CAT GTG TGG TGC ATC | 5540–5560 | 754 |
| DWV10R | GTG TAC GCT CCT TAA ATG CCT | 6294–6274 | |
| DWV11F | AATCAGCGCTTAGTGGA | 6249–6265 | 636 |
| DWV11R | ATCAGTCAACGGAGCATAC | 6866–6884 | |
| DWV12F | GCR TGA ACG TTC ATC TTC AAC | 6752–6772 | 720 |
| DWV12R | AAT CTA TGG ATT CTA GGT GCC | 7471–7451 | |
| DWV13F | TCACCAGGAATGGCAA | 7395–7411 | 746 |
| DWV13R | ATC CTT CAG TAC CAG CAA CA | 8140–8121 | |
| DWV14F | CATGTTGCTGGTACTGAAGGA | 8109–8129 | 483 |
| DWV14R | TCCAGGCACACCACATACAGC | 8571–8591 | |
| DWV15F | GTGTGCCTGGWTTAGATGGG | 8581–8600 | 622 |
| DWV15R | GCT AAR ATC TCT TGC GCC AT | 9202–9183 | |
| DWV16F | GATTCTGATGTTGCAGCTTC | 9081–9100 | 618 |
| DWV16R | CCGAATGCTAACTCTAGCGC | 9698–9678 | |
| DWV17F | GCATCCAACTAGACCCGTGT | 9576–9595 | 532 |
| DWV17R | AGGACGCATTACCACTAGTTGA | 10085–10107 | |

**Notes:**
Nucleotide positions refer to the published complete DWV genome sequence with GenBank accession number AY292384.
F, forward primer; R, reverse primer.

## Sequence and phylogenetic analyses

To research the DWV genome sequence characteristics of the two novel strains, 20 complete genome sequences of DWV were obtained online from the National Center for Biotechnology Information (http://www.ncbi.nlm.nih.gov/BLAST/) (Table 2).

**Table 2  DWVs strains used in this study.**

| No. | Name in this study | Accession number | Geographic origin | Length (nucleotides) | Host species | Submitted year |
|-----|--------------------|------------------|-------------------|----------------------|--------------|----------------|
| D1 | Italy-2002 | AJ489744 | Italy | 10,140 | *Apis mellifera* | 2002 |
| D2 | USA-2003 | AY292384 | USA | 10,135 | *Apis mellifera* | 2003 |
| D3 | USA-2015 | KT004425 | USA | 10,137 | *Apis mellifera* | – |
| D4 | France-2016 | KX373899 | France | 10,143 | – | 2016 |
| D5 | UK-2009 | GU109335 | UK | 10,140 | *Apis mellifera* | 2009 |
| D6 | Italy-2017 | KY909333 | Italy | 10,104 | *Vespa crabro* | 2017 |
| D7 | Chile-2012 | JQ413340 | Chile | 10,140 | – | 2012 |
| D8 | Austria-2016 | KU847397 | Austria | 10,164 | *European honeybee* | 2016 |
| D9 | China-2017 | MF036686 | China | 9,838 | *Apis mellifera* | 2017 |
| D10 | KV-2001 | AB070959 | Japan | 10,152 | *Apis mellifera* | 2001 |
| D11 | Korea-2012 | JX878305 | Korea | 10,096 | *Apis mellifera* | 2012 |
| D12 | Korea-2012 | JX878304 | Korea | 10,094 | *Apis mellifera* | 2012 |
| D13 | VDV-2004 | AY251269 | Netherlands | 10,112 | *Varroa destructor mites* | 2004 |
| D14 | Belgium-2016 | KX783225 | Belgium | 10,112 | *Apis mellifera* | 2016 |
| D15 | France-2016 | KX373900 | France | 10,103 | – | 2016 |
| D16 | UK-2010 | HM067438 | UK | 10,127 | *Apis mellifera* | 2010 |
| D17 | UK-2015 | KT215905 | UK | 10,264 | *Apis mellifera* | 2015 |
| D18 | UK-2014 | KJ437447 | UK | 10,140 | *Apis mellifera* | 2014 |
| D19 | UK-2010 | HM067437 | UK | 10,126 | *Apis mellifera* | 2010 |
| D20 | UK-2015 | KT215904 | UK | 10,264 | *Apis mellifera* | 2015 |
| D21 | China1-2017 | MF770715 | China | 10,141 | *Apis mellifera* | 2017 |
| D22 | China2-2018 | MH165180 | China | 10,105 | *Apis cerana* | 2018 |

Multiple alignments of nucleotides and amino acid analyses were performed using the ClustalW program in the MegAlign software program (DNASTAR, Madison, WI, USA). At the same time, the conserved domains of the nonstructural proteins in the C-terminal portion of the polyprotein were analyzed with reference to the functional domains of mammalian picornaviruses. To investigate the genetic variation and evolutionary characteristics of the isolates, DWV complete genome sequences of the classic Type A and Type B strains from 2001 to 2018 and two closely related viruses, KV (*Fujiyuki et al., 2004*) and VDV-1 (*Ongus et al., 2004*) in the GenBank database were selected for further analysis, including four isolates in Asia, 13 in Europe, three in America, and one in Oceania. Then, phylogenetic trees of ORF, VP1 and 3C+RdRp were reconstructed by the maximum likelihood method using MEGA5. The best-fitting nucleotide substitution model (GTR+I+G) was used for the all alignment datasets, which was determined by the lowest BIC score in MEGA 5.0 (*Tamura et al., 2011*). A total of 1,000 replicates of bootstrap resampling were used to ensure the reliability of individual nodes in each phylogenetic tree.

## Recombination analysis

Recombination analysis was performed using eight full-length genomes of DWV from Japan (KV-2001, AB070959), Korea (Korea I-2012, JX878304; Korea II-2012, JX878305),

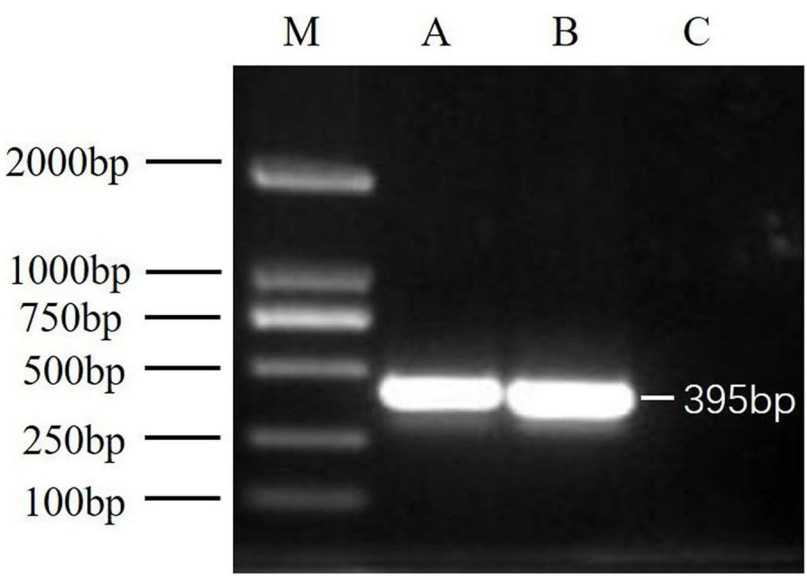

**Figure 1 RT-PCR detection results.** M, DNA marker DL2000; A, China1-2017 detection results; B, China2-2018 detection results; C, negative control (Healthy worker bee cDNA).

China (China-2017, MF036686; China1-2017, MF770715; China2-2018, MH165180), England (VDV-2013, KC786222), and the Netherlands (VDV-2004, AY251269), and the detection of inter-strain recombination, identification of closest parental sequences and localization of possible recombination break points were assessed using RDP4 software, which comprises RDP, GeneConv, Bootscan, MaxChi, Chimaera, SiScan, and 3SEQ algorithms. The standard settings of these algorithms were used with the default values of RDP4. The likelihood of recombination events was significant in at least four algorithms at $P < 1.0E$-6 or recombination consensus scores (RCS) above 0.6, based on the RDP4 analysis. Recombination events were considered possible when the $P$-value of at least three algorithms was below 0.05, and the RCS was between 0.4 and 0.6, and the likelihood of recombination events was considered insignificant when the RCS was under 0.4 with $P < 0.05$ (*Wang et al., 2015*; *Lee et al., 2017*; *Gao et al., 2018*). Moreover, the recombination events were further verified by GARD implemented in the Datamonkey web interface (*Delport et al., 2010*) and the credibility of the recombination breakpoints was assessed by the KH test.

### Nucleotide sequence accession number

The DWV nucleotide sequences of the strains China1-2017 and China2-2018 are accessible on GenBank (accession number: MF770715 and MH165180).

## RESULTS

### RT-PCR detection of DWV samples

After virus isolation, specific primer pairs were used to detect DWV in samples from Jinzhou and Qinhuangdao by RT-PCR, and fragments of approximately 395 bp were produced (Fig. 1). After sequencing and alignment, the nucleotide sequence identity exceeded 97% by
BLAST; thus, we successfully isolated two novel DWV strains from *A. mellifera* and *A. cerana* in China.

## Nucleotide sequence analysis of DWV isolates

The nucleotide sequences of China1-2017 and China2-2018 were 10,141 and 10,105 nt, respectively. The whole nucleotide sequences of China1-2017 and China2-2018 are enriched in A/U (China1-2017: A-29.17%, U-32.17%, G-22.76%, C-15.90%; China2-2018: A-29.09%, U-32.00%, C-16.16%, G-22.74%). The isolates contained a single major ORF from the 5′–3′ end, which was composed of 8,682 nt (ORF position in China1-2017: 1,146–9,827 bp; China2-2018: 1,135–9,816 bp), encoding 2,894 amino acids. Multiple sequence comparisons showed that the sequences of China1-2017 and China2-2018 were similar to those of previously reported DWVs/VDV-1 strains. Furthermore, compared to other DWV isolates, the nucleotide sequence identity and deduced amino acid sequence identity of China1-2017 and China2-2018 ranged from 84.3% to 96.7% and 94.8% to 98.6%, and 84.3% to 96.7% and 94.7% to 98.4%, respectively (Table S1).

## Amino acid sequence analysis of DWV isolations

As previously reported for mammalian picornaviruses, the ORF of China1-2017 and China2-2018 strains encoded a 2,894-amino-acid polyprotein, with structural proteins at the N-terminus and non-structural proteins at the C-terminus (*Lanzi et al., 2006*; *Organtini et al., 2016*). The structural and non-structural proteins were positioned in the genomes as follows: VP3 (China1-2017: 1,800–2,537 nt, China2-2018: 1,789–2,526 nt), VP1 (China1-2017: 2,601–3,848 nt, China2-2018: 2,590–3,837 nt), VP2 (China1-2017: 3,849–4,622 nt, China2-2018: 3,838–4,611 nt), helicase (China1-2017: 5,010–6,428 nt, China2-2018: 4,999–6,417 nt), and 3C-RdRp-protease (China1-2017: 7,686–9,827 nt, China2-2018: 7,675–9,816 nt). Moreover, six conserved domains in the helicase, 3C-pro, and RdRp were identified from China1-2017 and China2-2018. Three conserved helicase regions were found in the deduced amino acid sequences of the China1-2017 and China2-2018 ORF, ranging from 1,472 to 1,575, namely, domain A ($^{1472}$GxxGxGKS$^{1479}$), domain B ($^{1518}$Qx$_5$DD$^{1525}$), and domain C ($^{1561}$KKx$_4$Px$_5$NTN$^{1575}$). The 3C-pro conserved domains included the cysteine protease motif ($^{2305}$GxCG$^{2308}$) and the putative substrate-binding motif ($^{2322}$GxHxxG$^{2327}$). The highly conserved RdRp region, $^{2495}$TSxGxP$^{2500}$, was recognized between the deduced amino acid positions 2,495 and 2,500.

## Phylogenetic relationships of the DWV isolates

To assess the genetic relationships of the DWVs, three phylogenetic trees were constructed based on the VP1, 3C+RdRp segments, and ORF gene sequences. The results showed that the ORF and VP1 groups better explained the geographical distribution of DWV, and the 3C+RdRp-coding region better explained the genotype and diversity of DWVs.

In the ORF gene phylogenetic tree, the 24 DWV isolates (including China1-2017 and China2-2018) were divided into two groups (lineage A and B; Fig. 2A). The first group contained two master lineages (lineage A1 and A2), one of which included eight isolates from America and Europe; the second lineage included six isolates from Asia. China1-2017

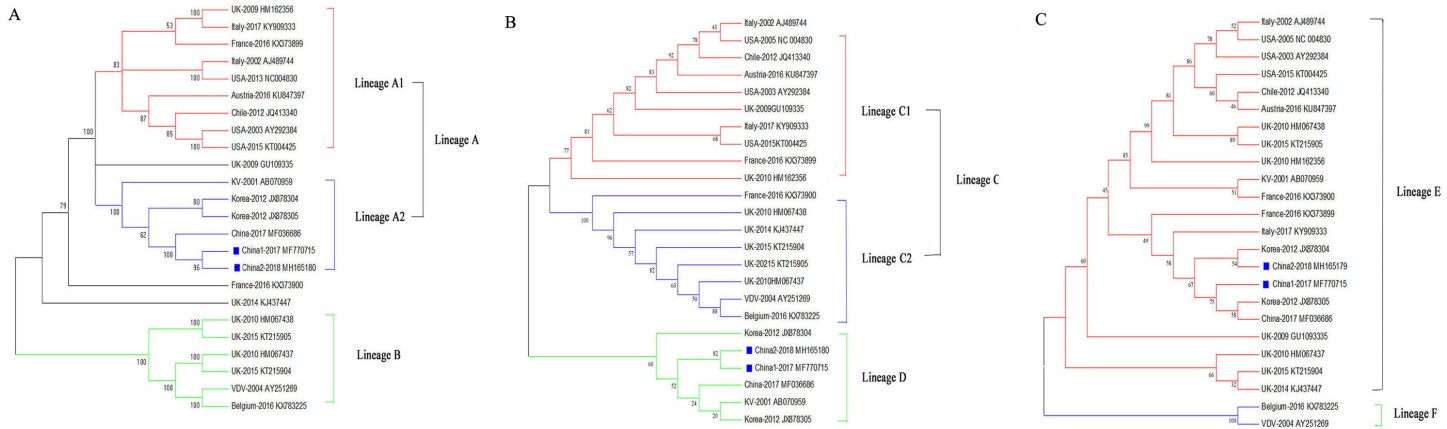

**Figure 2 Phylogenetic trees of DWV isolates.** (A) Phylogenetic tree based on the ORF-coding nucleotide sequence of DWV. (B) Phylogenetic tree based on the VP1 segment of DWV. (C) Phylogenetic tree based on the 3C+RdRp segment of DWV. All phylogenetic trees were constructed by maximum-likelihood method (ML) method with bootstrap resampling (1,000 replicates). The number at each branch of phylogenetic tree represents the bootstrap value (1,000 replicates). The different colors triangles indicate the different clusters of DWV isolates.

and China2-2018 belonged to lineage A2. Lineage B contained six isolates from Europe. We found that the phylogenetic tree of the ORF among DWV isolates correlated with the geographical distribution.

The phylogenetic tree based on the VP1 segment produced two groups (lineages C and D; Fig. 2B) and was similar to the ORF tree (Fig. 2A). The first group (lineage C) was further divided into two sub-groups (lineages C1 and C2): ten isolates from America, Europe, and Oceania formed lineage C1, whereas eight isolates formed lineage C2. The second group (lineage D) contained six isolates from China, Korea, Japan, and Europe.

The phylogenetic tree based on the 3C+RdRp segments produced two distinct groups (lineages E and F; Fig. 2C). The 22 isolates in lineage E belonged to DWV Type A, which included variants from America, Europe, Oceania, and Asia. Lineage F belonged to the classic DWV Type B, which only contained two isolates from Belgium and the Netherlands.

## Recombination analysis of the DWV isolates

To explore potential recombination signals in the DWV isolates from Asia, recombination signals were assessed using the RDP4 software. Using seven algorithms, nine recombination events were detected in Asian strains (Fig. 3B; Table 3). In all potential recombination events, three recombination events (events 2, 3, and 5) had a high degree of certainty based on the RDP4 software standard (Table 3). However, GARD analyses indicated that only one isolate (event 3), China2-2018, was identified as a recombinant at the breakpoint in the positions 4,266 and 7,507 nt with a high level of confidence (LHS, RHS *P*-values < 0.01). Based on the above analysis, event 3 was identified as the real recombination event. In event 3, the recombination of strain China2-2018 was located at the 4,266–7,507 nt region, with Korea I-2012 as an infer unknown parent and China-2017 as a minor parent (Figs. 4A–4C), which spanned the entire helicase ORF (Fig. 3).

                                                    

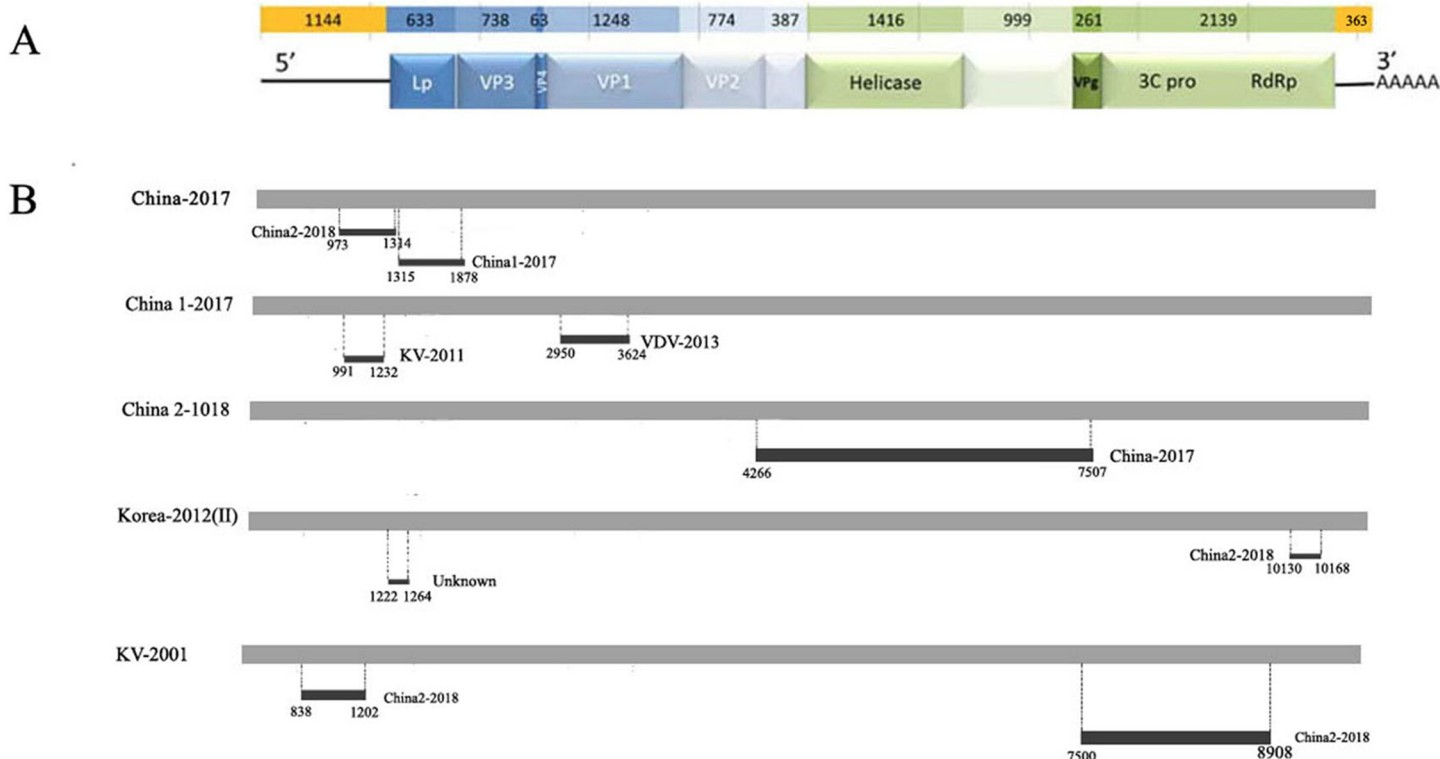

**Figure 3 Analysis of possible recombination in different DWV isolates from Asia.** (A) Genomic organization of DWV. (B) Summary of potential recombination in different isolates of DWV. Two events in China-2017 (Event 4, 5), three events in China1-2017 (Event 6, 7), one event in China2-2018 (Event 3), two events in Korea II-2012 (Event 1, 8), two events in KV-2001 (Event 2, 9). Dark bars in Fig. 4B indicating recombination regions with breakpoint positions and minor parent shown. Dot lines indicating breakpoints.

## DISCUSSION

Deformed wing virus is one of the most prevalent, pathogenic honeybee viruses in the world, and has been directly linked to colony collapse disorder (*Organtini et al., 2016*; *Kevill et al., 2017*). Despite the importance of DWV as a honeybee virus, only a limited number of complete genome sequences for DWV are available. In the present study, we determined the entire genome sequences for two DWV isolates collected from *A. mellifera* and *A. cerana* in China. Generally, DWV is only prevalent in *A. mellifera* and *V. destructor* and is not common in *A. cerana* (*Tentcheva et al., 2004*; *Xie, Huang & Zeng, 2016*; *Zhang & Han, 2018*). However, we obtained the complete genome sequence from the China2-2018 strain *A. cerana* for the first time in the present study, which helps to investigate the host range of DWV. The comparison of China1-2017 and China2-2018 showed 96.7% identity of nucleotide sequences and 98.1% identity of amino acids. Although the identity of the two isolates was relatively high, they did not belong to the same strain. The isolated strain sequences of similarities with 20 reference strains ranged from 84.3% to 96.4%, and the highest sequence identity was assigned to strain UK-2009 (GU109335), whereas Belgium-2016 (KX783225) and VDV-2004 had the lowest sequence identities. Therefore, the two novel isolates were significantly different from DWV Type B. The amino acid identity ranged from 94.8% to 98.6%, and the highest similarity belonged to the

**Table 3 Summary of possible recombination events in DWV isolates from Asian identified by RDP4.**

| Event number | Recombinant sequence (s) | Parental sequence (s) Major/Minor | Breakpoint position Begin/End | Recombinant score | P-Value for the seven detection methods in RDP4[a] | | | | | | |
|---|---|---|---|---|---|---|---|---|---|---|---|
| | | | | | R | G | B | M | C | S | T |
| 1 | Korea-2012(II) | KV-2001/ Unknown | 10130–10168 | 0.548 | 1.369$E$-7 | 1.513$E$-7 | NS | NS | NS | NS | 9.926$E$-4 |
| 2 | KV-2001 | Unknown/ China2-2018 | 7500–8908 | 0.621 | 5.316$E$-2 | NS | NS | 8.604$E$-7 | 2.184$E$-5 | 4.671$E$-3 | 3.816$E$-2 |
| 3 | China2-2018 | Unknow/ China-2017 | 4266–7507 | 0.49 | 5.309$E$-3 | 1.302$E$-7 | 2.551$E$-7 | 4.347$E$-8 | 5.820$E$-3 | 4.424$E$-12 | 3.675$E$-2 |
| 4 | China-2017 | China1-2017/ China2-2018 | 973–1314 | 0.393 | 8.010$E$-2 | NS | NS | 4.723$E$-4 | 4.14$E$-3 | 2.76$E$-2 | NS |
| 5 | China-2017 | KV-2001/ China1-2017 | 1315–1878 | 0.606 | 3.239$E$-7 | NS | 7.157$E$-6 | 4.662$E$-3 | 3.695$E$-3 | NS | NS |
| 6 | China1-2017 | China2-2018/ KV-2001 | 2950–3624 | 0.461 | NS | NS | NS | 6.260$E$-4 | 1.273$E$-3 | NS | NS |
| 7 | China1-2017 | KV-2001/ VDV-2013 | 991–1232 | 0.594 | 4.668$E$-3 | NS | NS | 3.228$E$-3 | NS | NS | NS |
| 8 | Korea-2012(II) | KV-2001/ China2-2018 | 1222–1264 | 0.410 | 1.217$E$-2 | 3.589$E$-1 | 1.330$E$-2 | 5.172$E$-3 | NS | 4.554$E$-4 | NS |
| 9 | KV-2001 | Korea-2012(I)/ China2-2018 | 838–1202 | 0.423 | 7.182$E$-4 | 1.959$E$-2 | 5.385$E$-4 | 1.031$E$-3 | NS | 1.357$E$-5 | NS |

**Notes:**
[a] Detection methods used in RDP4: R, RDP; G, GENECONV; B, BOOTSCAN; M, MaxChi; C, CHIMAERA; S, SISCAN; T, 3SEQ. Statistical significance is indicated according to the code described in "Materials and Methods."
NS, not significant.

strain VDV-2004 and the lowest to Chile-2012 (JQ413340). As shown in Table S1, virus strains from the same continents or from the same countries showed higher levels of identity of the nucleotides and amino acids, including the novel isolates; therefore, these viruses have been present in the honeybee populations for a long time and the viruses have evolved more or less independently. Moreover, six highly conserved motifs were identified from the two novel isolates, featuring the typical characteristics of iflaviruses.

Based on the phylogenetic analyses of ORF, VP1, and 3C+RdRp segments, the two novel isolates from China clustered within the same clade as other DWV strains form Asia; therefore, the novel isolates had a closer relationship with other Asian strains, and the DWV strains from Asia might have originated from the ancestor KV-2001 (AB070959). Furthermore, the phylogenetic analysis of DWV based on ORF and VP1 revealed numerous different geographically determined clades and the phylogenetic tree of VP1 also presented a clearer pattern regarding the geographical distribution; therefore, the strains with the closer relationship had different evolutionary rates in different environments and hosts. Moreover, the phylogenetic tree based on the VP1 sequences confirmed that China1-2017 and China2-2018 were more closely related to the isolates from Korea than to those from Japan. In RNA viruses, 3C-pro and RdRp genes tend to be highly conserved; therefore, 3C-pro and RdRp genes have usually been used to distinguish subtype classification of RNA viruses (*Baker & Schroeder, 2008*; *Geng et al., 2014*; *Kevill et al., 2017*). In the present study, the phylogenetic analysis of 3C+RdRp showed that
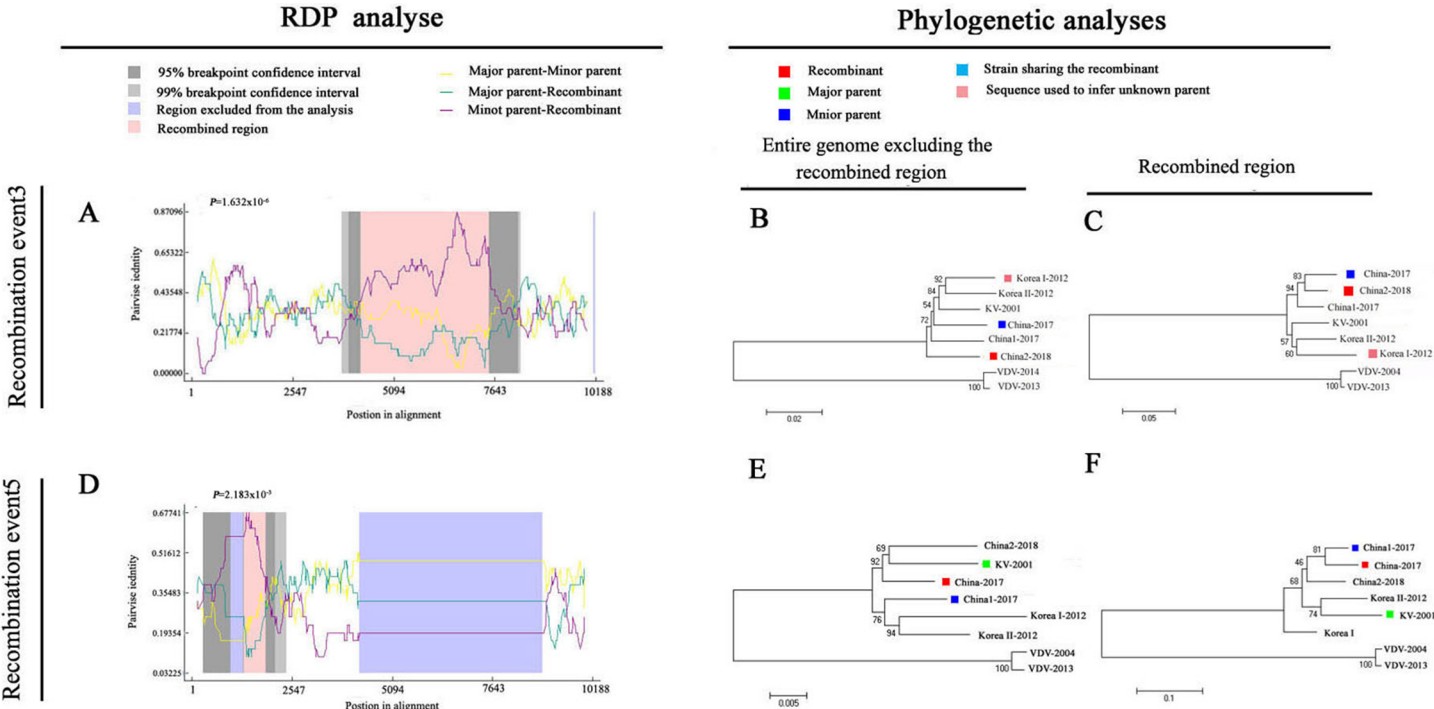

**Figure 4 Recombination analysis of full-length DWV genome sequences of China-2017 and China2-2018 isolations by RDP4.** An alignment of the eight DWV genomes from Asia and Europe was analyzed using the RDP4 software. Two potential recombination events were identified for Chinese strains (illustrated by A–F, respectively). The left part illustrates the results of RDP analyses (A and D). The right part presents phylogenetic analyses based on the full-length genome (excluding one of the terminal direct repeats) excluding the region of recombination (B and E) or based on the recombination region only (C and F) using UPGMA in MEGA5.0 with 1,000 replicates. Values on internal branches refer to the percentage of bootstrap replicates in which the branch was found; only values greater than 50% are shown. The scales illustrate the number of substitutions per nucleotide. The color code used is described at the top.

the two novel isolates all belonged to DWV Type A. With a difference in the ORF and VP1 tree, the China2-2018 (MH165180) strain was more closely related to the Korea-2012 strain (JX878304) than to the China1-2017 strain (MH770715). We suspect that the VP1 and 3C+RdRp genes had generated at a different evolutionary rate in different environments, which led to the diversity of the phylogenetic trees, because the structural proteins genes (VP1, VP2, and VP3) are more likely to mutate in the picornavirus (*Amin et al., 2014*; *Hu et al., 2016*).

Natural recombination is an important strategy for viruses to adapt to new environmental conditions and hosts. Recombination events have been observed in DWVs (*Seo et al., 2009*; *Lian et al., 2013*; *Dalmon et al., 2017*); however, the recombination events of Chinese DWVs have never been described. Based on geographical location, we used all DWV strains from Asia, the classic DWV Type B and two novel isolates from China in the present study to analyze the potential recombination events using RDP4 software and GARD, and the recombination event of the China2-2018 strain was confirmed (Table 3; Fig. 4). In this recombination event, the strains from China and Korea were mainly involved, including the novel isolates; therefore, the DWV recombination widely exists in honeybee colonies of East Asia. In general, irregular and complicated

recombination patterns indicate that the recombination events are usually random, although a detailed understanding of the mechanism involved in such recombination phenomenon must be clarified. When compared with the DWV recombinant isolated by Dalmon et al., we found that all recombinant sites located in the region encoded non-structural proteins. The apiculture characteristic of China is probably the greatest factor that led to the DWV epidemic in *A. cerana* colonies. In China, there are a large number of *A. mellifera* and *A. cerana* colonies found in the same region during the nectar collecting season, which could promote the transmission of DWV from *A. mellifera* to *A. cerana* by pollen. In this interaction among the virus, host, and infectious vector, the recombination strains continuously appear. Future studies are required to discover more recombination phenomenon and the occurrence of recombination events may contribute to the high levels of genetic diversity and viral adaptability to host in DWVs, which may increase the potential of this virus to threaten successful beekeeping.

## CONCLUSIONS

In summary, we found two novel DWV isolates from China and reported the first complete genome sequence of DWV from *A. cerana*. Based on phylogenetic trees, the novel DWV isolates from China were confirmed to be the closest related to the strain from Korea. Furthermore, the recombinant phenomenon was discovered in the novel isolates of China2-2018, which is the first description of DWV recombination in China. Our study has not only revealed the presence of novel DWV recombinants in China but also provides information that may be useful for further research on the phylogenetic origins of Chinese DWV strains.

### Funding

This work was supported by grants from the National Natural Science Foundation of China (31772760), the Award for "Liaoning Distinguished Professor", the Liaoning Province Natural Sciences Foundation of China (20170540377 and 20180550289), and the National College Students' innovation and entrepreneurship training program of Jinzhou Medical University (201710160000189). The funders had no role in study design, data collection and analysis, decision to publish, or preparation of the manuscript.

### Grant Disclosures

The following grant information was disclosed by the authors:
National Natural Science Foundation of China: 31772760.
"Liaoning Distinguished Professor", the Liaoning Province Natural Sciences Foundation of China: 20170540377 and 20180550289.
National College Students' innovation and entrepreneurship training program of Jinzhou Medical University: 201710160000189.

## Competing Interests

The authors declare that they have no competing interests.

## Author Contributions

- Dongliang Fei conceived and designed the experiments, analyzed the data, authored or reviewed drafts of the paper.
- Yaxi Guo performed the experiments, prepared figures and/or tables.
- Qiong Fan contributed reagents/materials/analysis tools.
- Haoqi Wang performed the experiments.
- Jiadi Wu performed the experiments.
- Ming Li performed the experiments.
- Mingxiao Ma conceived and designed the experiments, analyzed the data, authored or reviewed drafts of the paper, approved the final draft.

## Animal Ethics

The following information was supplied relating to ethical approvals (i.e., approving body and any reference numbers):

This research was approved by the Experimental Animal Ethics Committee of Jinzhou Medical University (No. 20180016).

## Field Study Permissions

The following information was supplied relating to field study approvals (i.e., approving body and any reference numbers):

The collection of sample was approved by the agricultural and rural comprehensive service center of Jinzhou (No. 20180619) and the local apiaries.

## DNA Deposition

The following information was supplied regarding the deposition of DNA sequences:

The DWV nucleotide sequences of the strains China1-2017 and China2-2018 are accessible at GenBank: MF770715 and MH165180.

## Data Availability

Data is available at GenBank, accession numbers MF770715 and MH165180.

## Supplemental Information

Supplemental information for this article can be found online at http://dx.doi.org/10.7717/peerj.7214#supplemental-information.

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
