# Peer review of "Phylogenetic and recombination analyses of two deformed wing virus strains from different honeybee species in China"

_PeerJ, doi:10.7717/peerj.7214_

## Round 0.1 · original submission · Major Revisions

Dear Prof. Ma,

Thank you for your submission to PeerJ.

As reviewers' comments, I also suggested that more details are required in the description of the methods, including phylogenetic and recombination analyses.

Two reviewers gave good suggestions. I believe those suggestions will improve your paper.

With kind regards,

Jia-Yong Zhang
Academic Editor, PeerJ

Reviewer 1 ·

Basic reporting

It meets standards of English, article structure.

Experimental design

The research is original. It appears to be rigorous, but more details are required in the description of the methods, including phylogenetic and recombination analyses.

Validity of the findings

Some recombination analyses in this study are insufficient to assess the robustness of the results, although the descriptions of the methods of recombination analyses can be amended by improving the text. Therefore, it needs substantial additional work, including some distinguishing biological property for the potential recombinants.

Additional comments

In this study, Fei et al. conduct phylogenetic and recombination analyses to investigate the evolutionary relationships of the deformed wing virus. I value the efforts that the authors have put into this study, but it is not suitable for publication at the present format. By accessing the results presented in this paper, I don't think in silico recombination analysis is sufficient to justify recombination events. In Figure 3, no putative recombination breakpoints were observed in the China1-2017 or China2-2018 strains from SimPlot analyses. Simultaneously, the recombination events were detected in the two strains by RDP packages without high confidence (at least supported by four but not three different algorithms with an associated P value of < 1.0E-6) in Table 4. To avoid false identification, the settings of key parameters in RDP packages should have to be modified according to the size of the sequence dataset before identifying them as recombinants. In addition, the authors would have to show that these chains have some distinguishing biological property. This does not appear to be the case.

Some minor comments are listed as follows:

ABSTRACT
Line 33 Please replace ‘Provinces’ with ‘Cities’

Line 33-34 ‘MF770715’and ‘MH165180’ should be ‘accession number MF770715’ and ‘accession number MH165180’, respectively

Line 49-50 ‘similarity’ should be ‘identity’

Line 157, 169 and 180
Please replace the terms ‘GenBank accession no’ with ‘accession number’ throughout the manuscript.

Line187-190
It is now generally accepted that model-based phylogenetic algorithms, such maximum likelihood (ML) or Bayesian inference (BI) produce more reliable results than distance-based (i.e. neighbor joining, NJ) and parsimony (maximum parsimony, MP) methods. Therefore, it would be preferred to reconstruct the phylogenetic tree using either BI or ML algorithms. Notably, which substitution model and what criterion providing the best fit to the data in this study should be mentioned.

Line191-212
For recombination analyses, the potential recombinants and parental strains were firstly identified using seven methods implemented the RDP v4 packages, and then further analysed using Simplot or Bootscan implemented Simplot v3.5. In addition, multiple different approaches were recommended to investigate and confirm the recombination events. For example, phylogenetic networks could be employed when reticulate events such as recombination. Besides, GARD implemented in the HyPhy packages could be used to further confirm the tentative recombinants.

Line 220
Please replace ‘homology’ with ‘identity’ thorough the manuscript. In addition, it is unclear that the best sequence threshold criterion for this genus species demarcation. This should be mentioned in the text.

Reviewer 2 ·

Basic reporting

Please see my 'General comments for the author'.

Experimental design

Please see my 'General comments for the author'.

Validity of the findings

Please see my 'General comments for the author'.

Additional comments

In this manuscript, the authors report complete genome sequences and phylogenetic analyses of two deformed wing virus (DWV) strains, China1-2017 and China2-2018, which were isolated from Apis mellifera and Apis cerana colonies kept in China, respectively. Although I agree that most of the experimental designs and data presented in this manuscript sound solid, I think that the authors need to add some discussions about the results obtained in this study in the Discussion section. I have some major and minor comments, as follows:

Major comments:
1) Although the authors state in their Abstract that ‘The sequences were compared with…sequences of two closely related viruses, Kakugo virus (KGV) and Varroa destructor virus-1 (VDV-1).’, these results are completely missed in their Results and Discussion sections. These results need to be included in Tables 2-4, and Figs. 2-5 and discussed in their Discussion section.
2) My impression is that most of the current Discussion section is a mere repetition of information already described in the Introduction and Results sections, and does not contain necessary discussion based on their findings. For example, the authors need to discuss possible reasons why the results of phylogenetic tree of the ORF and VP1 groups better explained the geographic distribution, while the results of the 3C;RdRp-coding region better explained the genotype and diversity of DWVs.
3) The authors also need to discuss whether DWV strains are infectious to both Apis mellifera and Apis cerana in their Discussion section, based on their results of recombination analysis.
4) Please explain why it is so far thought that ‘Generally, DWV is only prevalent in A. mellifera and V. destructor, and not common in A. cerana’ and discuss why different results were obtained in this study.

Minor points:
1) Ls. 220-221; Although the authors state ‘The nucleotide sequence homology…BLAST alignment (Fig.1)’, Fig.1 shows the results of gel electrophoresis but not the results of sequence homology analysis.
2) Ls. 225-228; ‘each of which contains a single…’ is correct?
3) L. 292; ‘52%’ is correct?
4) L. 300; (Table 3) is correctly (Table 4)?
5) Table 1; DWV10 ‘6294-9274’ seems incorrect.
6) Table 3 Note; Table 3 contains no ‘shaded frames’.
7) Table 4 Note; What does ‘2’ that precedes ‘NS: not significant’ mean?
8) Table 4; Please explain the reason why the results of event number 1 and 6 are shown with bold letters.
9) Fig. 1; What does ‘C=Negative control’ mean?
10) Fig. 4; Please explain the meanings of green, red and blue triangles in the legend.
11) Fig. 5; Please consider to avoid overlapping of vertical lines and some words colored in red. Also consider to show the meanings of red, blue and black lines in the figure legend but not in an inlet (they are too small).

---

## Round 0.2 · Major Revisions

Dear Prof. Ma,

Thank you for your resubmission to PeerJ.

Although only one review is back, that reviewer suggested "Major revisions". I suggest you should carefully revise the manuscript according to the reviewers.

More detailed description in methods should be added to make more readers understand. I also suggest you should improve your English grammar and expression.

With kind regards,

Jia-Yong Zhang
Academic Editor, PeerJ

Reviewer 1 ·

Basic reporting

It meets standards of article structure, but the language needs some work. This can be improved with extensive language editing.

Experimental design

The research is original. It seems to be rigorous, but more details are clarified in the description of the methods.

Validity of the findings

Some statistical tests are not described in sufficient detail to assess the robustness of some of the results.

Additional comments

– This is a revised version of the paper by Fei et al. on Phylogenetic and recombination analyses of DWV in China. I think the authors made some efforts to improve the manuscript that now looks much better. I am not a native English speaker, but I think that language should be checked to correct several typos and/or sentences which are not clear to me.
After going through the authors' responses, I have the following comments:

METHODS
– Line 192 and 203-208 Please delete the relevant contents of similarity analyses to avoid confusion
– Line 199 replace "substitutionmodel" by " substitution model"
In addition, I think it would be worth specifying the criterion used to select the substitution model for phylogenetic reconstruction.

RESULTS
– The authors mention that some potential recombination signals were detected by RDP package. In fact, Table 4 shows only a significant recombination signal was observed in the China2-2018 (event 3) strain according to the threshold used in RDP4, as described in the method section. However, the authors found no evidence of recombination in the data set using Simplot with reference strains, because no significant recombination breakpoints are shown in Figure 3. The results of Simplot analysed do not support the results of recombination analysis using the RDP4 software. The conclusion on recombination analyses is not only confusing but also overstated based on the results mentioned above. Although the authors answered to my comment on this contradiction by descripting that “it's not appropriate to put the SimPlot analysis and the recombination analysis together”. Unfortunately, this is not a convincing explanation on this contradiction. Thus, I am basically submitting the same review than I previously did. To avoid unnecessary confusion, in my opinion, the relevant contents about the Simplot analyses would be removed from the manuscript.
– In addition, the authors should only mention the recombination events for which the detection of potential recombinants are significant whereas should not mention any non-significant results, which are meaningless.

– Line 241 replace "isloates" by " isolates"
– Line 307 replace "alorithms" by " algorithms"
– Line 308 replace "potenial" by " potential"
– Line 313 replace "recombina" by " recombination"
– Line 315 replace "evnet 5" by " event 5"

DISCUSSION
Briefly, for quantity of provided data, contents of discussion are unsatisfactory.

– Line 323 replace "In Addition" by "In addition"
– Line 323 replace "different different" by "different"
– Line 382, 385, 386 replace "honeybee clonies" by "honeybee colonies"?
– Line 382 replace "ofter" by after?
– Line 391 replace "recombinated phenomenon" by "recombination phenomenon"

Table 3
– This table is not particularly informative, it would fit better as supplementary.

Figure 2 figure legend
replace "Komura-2-parameter" by "Kimura-2-parameter "?

---

## Round 0.3 · Minor Revisions

Dear Prof. Ma,

Thank you for your submission to PeerJ.

According to one reviewer's suggestion, you should revise phylogenetic method especially in model selection.

I found that two authors is new in the revised manuscript. I require you should add the statement document as a supplemental file that all authors agree to add two authors.

I am looking forward your revised paper.

Best wishes,
Jia-Yong Zhang
Academic Editor, PeerJ

Reviewer 1 ·

Basic reporting

It meets standards of article structure, but there are still some typos in the revised manuscript although it has been proof-edited by FSeditor.

Experimental design

It appears to be rigorous, but minor details need to be clarified in the M&M section.

Validity of the findings

It is technically sound, leading to the conclusions listed in the manuscript.

Additional comments

The revised version of the manuscript by Fei et al. has improved compared with the previous version, but a few points might require some attention, especially in the revised manuscript with track changes (please see the attached).

Some minor comments and the typos that should be checked carefully are listed as follows:

-Abstract/Line 62 'fromin'? What does this mean?

-Line 68 'decreaded' change to 'decreased'

-Line 93 '…areatrophy of the wings' change to'…are atrophy of the wings'

-Line 98 'hasbeen distributed' …'change to 'has been distributed …'

-Line 127 'usingrecombination' change to 'using recombination'

-Lin 204-208 Which the nucleotide models and what criterion were selected for each dataset (ORF, VP1 and 3C+RdRp) to perform maximum likelihood (ML) analyses, respectively? The GTR+G+I and Kimura-2-parameter models were mentioned in the manuscript. Do the authors mean that the GTR+G+I model was used for three datasets? If that’s the case, please rephrased this sentence. Here are for your reference. “Then, phylogenetic trees of ORF, VP1 and 3C+RdRp were reconstructed by the maximum likelihood method using MEGA5. The best-fitting nucleotide substitution model (GTR+I+G) was used for the all alignment datasets, which was determined by the lowest BIC score in MEGA 5.”

-Line 208 Please remove the citation of “Kurmar et al, 2016” since it was not referenced by the authors in the manuscript.

-Line 246 'isloates' change to 'isolates'

-Line 389 'related to to' change to 'related to'

-Line 390 'wasdiscovered' change to 'was discovered'

-Line 449-450 Please remove the reference of 'Kumar S, Stecher G, Tamura K. 2016.MEGA5: Molecular Evolutionary Genetics Analysis Version 7.0 for BiggerDatasets. Mol Biol Evol.' because the citation is wrong. In fact, this is a MEGA 7 citation, not MEGA5.

-Figure 2 Legend please remove 'and the Kimural-2-parameter method'

Annotated reviews are not available for download in order to protect the identity of reviewers who chose to remain anonymous.

Reviewer 2 ·

Basic reporting

no comment

Experimental design

no comment

Validity of the findings

no comment

Additional comments

The authors have made much efforts to appropriately revise their manuscript, and I think the revised version has drastically been improved when compared with the previous one.

The results reported in this manuscript are novel and important.

I am satisfied with all of the authors responses to my and other Reviewers' comments and have no more concerns to this manuscript.

---

## Round 0.4 · Minor Revisions

Dear Dr. Ma,

Thank you for your submission to PeerJ.

There are still a few problems which need to revise.

P4L83, P8L198, P9L219, P9L220, P9L230, P11L289. You should add blank in some words and add the reference which was not found in the References. So you also should check all references in the text and in the References.

I attached the PDF file.

With kind regards,
Jia-Yong Zhang
Academic Editor, PeerJ

---

## Round 0.5 · accepted · Accept

Dear Prof. Ma,
Thank you for completing the revisions and congratulations on the acceptance of the manuscript.

With kind regards,
Jia-Yong Zhang
Academic Editor, PeerJ

#